# Exploring the Use of Medicinal Plants and Their Bioactive Derivatives as Alveolar NLRP3 Inflammasome Regulators during *Mycobacterium tuberculosis* Infection

**DOI:** 10.3390/ijms22179497

**Published:** 2021-08-31

**Authors:** Nontobeko E. Mvubu, Thamsanqa E. Chiliza

**Affiliations:** School of Life Sciences, College of Agriculture, Engineering and Science, University of KwaZulu-Natal, Westville 3630, South Africa; chilizat@ukzn.ac.za

**Keywords:** NLRP3 inflammasomes, alveolar macrophages, pulmonary epithelial cells, medicinal plants, medicinal plant derivatives, *Mycobacterium tuberculosis*, immunopathology, interleukin 1 cytokines

## Abstract

*Mycobacterium tuberculosis*, the causative agent of tuberculosis (TB), is a successful intracellular pathogen that is responsible for the highest mortality rate among diseases caused by bacterial infections. During early interaction with the host innate cells, *M. tuberculosis* cell surface antigens interact with Toll like receptor 4 (TLR4) to activate the nucleotide-binding domain, leucine-rich-repeat containing family, pyrin domain-containing 3 (NLRP3) canonical, and non-canonical inflammasome pathways. NLRP3 inflammasome activation in the alveoli has been reported to contribute to the early inflammatory response that is needed for an effective anti-TB response through production of pro-inflammatory cytokines, including those of the Interleukin 1 (IL1) family. However, overstimulation of the alveolar NLRP3 inflammasomes can induce excessive inflammation that is pathological to the host. Several studies have explored the use of medicinal plants and/or their active derivatives to inhibit excessive stimulation of the inflammasomes and its associated factors, thus reducing immunopathological response in the host. This review describes the molecular mechanism of the NLRP3 inflammasome activation in the alveoli during *M. tuberculosis* infection. Furthermore, the mechanisms of inflammasome inhibition using medicinal plant and their derivatives will also be explored, thus offering a novel perspective on the alternative control strategies of *M. tuberculosis*-induced immunopathology.

## 1. Introduction

*Mycobacterium tuberculosis* has proven to be a successful pathogen over the years, responsible for an estimated 1.2 million deaths, with 10 new million cases annually. This opportunistic pathogen is also accountable for another 0.2 million estimated deaths among HIV-TB co-infected individuals [1]. The success of *M. tuberculosis* as a human pathogen is due to the emergence, persistence and transmission of genetically diverse drug resistant strains in different geographic locations [2], delays in diagnosis [3,4], and failure of the individual’s immune system to contain and eliminate the pathogen [5].

In the initial stages of pulmonary TB infection, physical barriers and innate immune response are present to prevent and control the infection. Initial interaction between the *M. tuberculosis* bacillus and the alveolar lining present an opportunity for containment of infection by innate cells, such as type I and type II pulmonary epithelial cells [6], resident alveolar macrophages, and surrounding dendritic cells and neutrophils [7,8]. The highly abundant pulmonary epithelial cells in the alveolar lining are connected by tight junctions, adherens junctions, and desmosomes [9,10] that are relatively impermeable [11]. Residence alveolar macrophages are ideally located to render microbicidal action through the activation of phago-lysosome against invading bacilli [12]. Surrounding dendritic cells and neutrophils link the innate and adaptive immune responses and produce toxic mediators to eliminate the invading bacilli, respectively [13,14]. *M. tuberculosis* is a successful pathogen due to its ability to bypass the host innate defence mechanisms, which eventually leads to the granuloma formation [15].

Innate cells express a range of host pathogen recognition receptors (PRRs) that interact with pathogen associated molecular patterns (PAMPs) in *M. tuberculosis* [16,17,18]. Interaction of the *M. tuberculosis* cell surface lipid antigens with the host [18] innate cells receptors such as TLR4 stimulate the activation of the NLRP3 inflammasome pathway through the Myeloid differentiation primary response 88 (MyD88) adaptor molecule [19,20]. Activation of the NLRP3 inflammasome is critical in the production of inflammatory cytokines of the IL1 family [21,22]. Several studies have shown that inflammasome activation and early production of IL1 cytokine family contribute to the early immune response to *M. tuberculosis*, thus, offering an effective control strategy [23,24]. Paradoxically, overstimulation of the inflammasome resulting in high production of IL1 cytokines has been shown to be detrimental to the host due to overstimulation of the immune system, leading to immunopathology in the host [25,26]. Thus, controlling overstimulation of the alveolar inflammasome provides an ideal target for prevention of the inflammatory response that is associated with immunopathology during TB infection.

Several studies have shown the effectiveness of medicinal plants and their bioactive derivatives as a promising approach to TB treatment since these are naturally sourced compounds with minimal side effects [27,28,29,30]. Medicinal plants and their respective active compounds exhibit their anti-inflammatory mechanisms through inhibition of the NLRP3 inflammasome-associated transcripts [31,32], inflammasome assembly [33,34] and production of inflammasome proteins, including IL1 cytokines [35] in in vitro and in vivo *M. tuberculosis* infection models. The current review reveals the molecular mechanism behind inflammasome activation on resident alveolar macrophages and pulmonary epithelial cells that results in IL1 cytokine production, which have been shown to contribute to the alveolar inflammatory response during *M. tuberculosis* infection. Furthermore, medicinal plants and their active derivatives that can be explored to prevent overstimulation of the alveolar NLRP3 inflammasome and their mechanisms of action will be reported as a novel host-directed therapy against *M. tuberculosis*.

## 2. *M. tuberculosis* Interaction with Alveolar Macrophages and Pulmonary Epithelial Cells

Initial interaction between *M. tuberculosis* PAMPs and mammalian PRRs is crucial for the bacilli to gain entry into the cells and initiate the pathogenesis process. Several studies have shown that macrophages express complement (CR) [36,37] and Fc gamma (Fcγ) receptors [38] during *M. tuberculosis* infection. Reduced *M. tuberculosis* bacterial uptake was observed in macrophages isolated from CR3-knockout mice compared to the wild type [39]. Moreover, several studies [40,41,42] have revealed an important role of CR receptors in opsonic and non-opsonic phagocytosis of *M. tuberculosis* in human macrophages. Initially, the Fcγ receptors were thought to contribute to *M. tuberculosis* killing because they mediate uptake of Immunoglobulin G (IgG)-opsonised mycobacteria [43], however, they were later shown to reduce the Th1 response by attenuating IL-12p40 production [38]. Collectively, these findings suggest that both CR and Fcγ receptors are involved in *M. tuberculosis* uptake in macrophages through opsonic and non-opsonic phagocytosis.

Mannose receptors (MR) are transmembrane C-type (calcium dependent) lectins that recognise and bind to the expressed mannose sugars in the lipoarabinomannan (LAM) of *M. tuberculosis* surface [44,45,46,47]. Interaction between the MR and *M. tuberculosis* mannosylated LAM promotes phagocytosis in macrophages [44], which may be dependent on virulence of the strain [47]. Other well characterised receptors that contribute to the update of *M. tuberculosis* in alveolar macrophages include Surfactant Protein A [48,49], Dectin-1 [50,51], NOD2 [52], and CD14 [53]. TLRs [54,55] are one of the well-known host PRRs that are expressed by alveolar macrophages in recognition of the diverse PAMPs in *M. tuberculosis* surface as excellently reviewed by Faridgohar and Nikoueinejad [18]. Despite the advances and knowledge generated on host PRRs expressed on alveolar macrophages, differential activation of these receptors by different lineages of *M. tuberculosis* remains to be investigated as this can contribute to better understanding of the inflammatory response induced by genetically diverse clinical strains.

Previously, not much was known about specific receptors expressed by pulmonary epithelial cells during *M. tuberculosis* infection. Bermudez and Goodman [56] initially proposed that *M. tuberculosis* use microtubules and microfilament pathways to gain entry into pulmonary epithelial cells. This was shown by significant reduction of intracellular bacilli when these pathways were blocked [56]. Lee et al. [57] revealed an upregulation of Dectin-1 receptors in a TLR-2 dependent manner of A549 pulmonary epithelial cells during *M. tuberculosis* infection. Through the production of a Dectin-1 receptor protein, they showed that these non-phagocytic cells might use this receptor for recognition of *M. tuberculosis*. The *M. tuberculosis* strains of East-African Indian and Euro-American lineages both induced down-regulation of the NOD1 receptor of the A549 pulmonary epithelial cells at 72 h post-infection [58]. However, a time course analysis of the NOD-associated receptors is needed to establish their involvement in the recognition of *M. tuberculosis*. Mannose and DC-SIGN receptors have not been linked to *M. tuberculosis* invasion of pulmonary epithelial cells and this area remains to be investigated.

TLRs are involved in the recognition of different *M. tuberculosis* cell wall-associated structures, such as acylated lipoproteins (TLR1, TLR2, TLR6), ESAT-6, mycolic acid (TLR2), acylated lipomannan (TLR4), and trehalose dimycolates (TLR3, TLR4, and TLR9), respectively [18]. Sequeira et al. [59] showed the role of TLR2 of pulmonary epithelial cells in the production of pro-inflammatory cytokines during infection by *M. tuberculosis*. Inhibition of TLR2-mediated response by mycolic acid and mce1 operon mutant significantly reduce the production of both IL8 and MCP-1, which suggests that TLR2 is essential to effective inflammatory response of pulmonary epithelial cells to *M. tuberculosis*. Previously, we showed an increased expression of TLR2, TLR3, TLR5, and TLR8 in pulmonary epithelial cells at 48 h post-infection by genetically diverse *M. tuberculosis* clinical strains of East-Asian and Euro-American lineages, leading to production of a diverse range of inflammatory cytokines. Upregulation of TLR4 was only induced by the Beijing (East-Asian) and F11 (Euro-American) strains, while TLR1, TLR6, TLR7, and TLR9 were downregulated by all strains [60].

A transcriptomic study performed by Hadifa et al. [58] revealed an increased expression of TLR1 and TLR3 by East-African Indian and Euro-American lineages of *M. tuberculosis* at 72 h post-infection of A549 pulmonary epithelial cells. Furthermore, the East-African Indian strain induced increased expression of TLR2, TLR5, and TLR9; while the Euro-American strain induced down-regulation of TLR4, TLR5, and TLR9. Both transcriptomic studies [58,60] indicate the TLRs are activated by different lineages of *M. tuberculosis* in pulmonary epithelial cells at 48 and 72 h post infection and these PRRs may be the main receptors used by this pathogen to gain entry into these cells.

Upon stimulation of the alveolar macrophages and pulmonary epithelial cells TLRs by *M. tuberculosis* PAMPs, the MyD88, TIR-domain-containing adapter-inducing interferon-β (TRIF), TRIF-related adapter molecule (TRAM), Toll-interleukin 1 receptor domain containing adaptor protein (TIRAP), B-cell adaptor for PI3K (BCAP), and sterile alpha and TIR motif containing (SARM) are adaptor molecules that are central components of the TLR signalling pathway [61]. Previously, we showed that genetically diverse strains of *M. tuberculosis* induced high enrichment of the TLR signalling pathways in pulmonary epithelial cells with increased expression of the MyD88 adaptor molecule [62]. Several studies have linked phagocytosis of *M. tuberculosis* by alveolar macrophages to the MyD88-signalling pathway [63,64] with a highly impaired inflammatory response in MyD88-knockout studies [65,66]. Thus, the activation of these adaptor molecules in pulmonary epithelial cells and alveolar macrophages results in downstream signalling cascade, leading to the activation transcriptional factors that are responsible for expression of inflammatory cytokines, including interferons and IL1 cytokine family [67]. Moreover, interaction between *M. tuberculosis* surface antigens with TLR4 triggers the NF-kB signalling pathway to induce expression of IL1β and IL18, whose maturation and production is dependent on the inflammasome assembly [68].

## 3. Activation of the Alveolar NLRP3 Inflammasome during *M. tuberculosis* Infection

Inflammasome is a multi-protein complex intracellular structure, which induces maturation of inflammatory cytokines, IL1β, and IL18 through the activation of caspase-1 and caspase-4/11 [69]. The activation of the inflammasome is triggered by signalling pathways that involves host PRRs, such as NOD-like receptors and TLRs [19]. Inflammasomes function in modulating host defence response, as well as pyroptosis, which is an inflammatory induced lytic mode of cell death that can be caspase-1 or caspase-11 dependent [70]. The assembly of an inflammasome is a coordinated signalling event, which is essential to producing an immune response after sensing *M. tuberculosis* [71,72]. The canonical NLRP3 inflammasomes is mediated by the activation of Caspase 1, while the non-canonical NLRP3 activation can be activated downstream of Caspase 4/11, respectively [19,67].

The canonical inflammasome pathway is initiated through the activation of caspase 1 by NLRP3, which responds to stimulation by IL1R, Tumor necrosis factor (TNF) or a TLR ligand that binds its cognate receptor, which results in the translocation of NF-κB into the nucleus. Thereafter, the expression of NLRP3 and pro-IL1β are induced, while pro-IL18 is constitutively expressed within cells. Several studies [18,54,55,58,60] have shown that *M. tuberculosis* activate TLR4 that initiate the canonical inflammasome in alveolar macrophages and pulmonary epithelial cells. Infection by genetically diverse clinical strains of *M. tuberculosis* revealed an upregulation of NFKB1 and NFKB2 transcriptional factors that lead to the transcription of IL1β, while IL18 was slightly down-regulated at 48 h infection by East-Asian and Euro-American lineages [60]. The Euro-American lineage strain induced upregulation of both NFKB1 and NFKB2 transcriptional factors at 72 h, while both transcripts were downregulated by the East-African Indian lineage strain, leading to downregulation of IL1β by both strains [58]. The *M. tuberculosis* Beijing strains of the East-Asian lineage were differentially recognised by TLR2 and TLR4 in macrophages resulting in distinct cytokine profiles, including IL1β [73], activating the canonical and non-canonical NF-κB signalling pathways [74]. Collectively, these findings indicate that *M. tuberculosis* interact with the TLR4, resulting in the activation of the NF-κB transcriptional factor, which is responsible for the transcription of pro-inflammatory cytokines, including IL1 by the alveolar macrophages and pulmonary epithelial cells. Furthermore, *M. tuberculosis* lineage-specific differences of the canonical inflammasome activation suggest that genetically diverse strains may induced diverse inflammasome patterns in the alveolar lining.

The second signal of the canonical inflammasome pathway involves the assembly of the NLRP3 inflammasome complex, which results in the recruitment of adaptor molecule, CARD and pro-caspase-1 in order to induce the processing and secretion of IL1β and IL18 cytokines. Pulmonary epithelial cells exhibit increased regulation of NLRP3 (by only F11 and H37Rv strains) PYCARD (ASC) and CASP1 during early infection by clinical strains of *M. tuberculosis* [60]. This upregulation may be time and lineage specific since the Euro-American lineage downregulated CASP1 at 72 h [58] in pulmonary epithelial cells. High levels of the mature IL18 were predominantly observed in *M. tuberculosis*-stimulated pulmonary epithelial cells, which indicated the upregulation of IL18 expression at both transcriptional and post-transcriptional levels, suggesting the involvement of CASP1 enzymatic activity [75]. There were no significant differences in the production of IL1β by pulmonary epithelial cells, which increased from 24, 48, and 72 h post-infection by wild type, *Mycobacterium tuberculosis* curli pili (MTP) mutant and complemented strains of *M. tuberculosis* [76], suggesting that IL1 cytokine production is not dependent on the MTP antigen. It should be noted that the activation of the canonical NLRP3 inflammasome pathway in pulmonary epithelial cells, if present, may be dependent on the lineage of the infecting strain, as well as the infection time as shown by the differences observed for the Euro-American and East-African Indian lineages [58,60]. Stimulation of the canonical inflammasome by *Mycobacterium* and it associated factors is well described and characterised in alveolar macrophages [72,77,78], as these cells have been shown to produce inflammatory IL1β and IL18 [77,78,79].

Non-canonical NLRP3 inflammasome mediated by CASP4/11 [80] has been shown to be activated by intestinal Gram-negative bacterial such as *Citrobacter rodentium*, *Escherichia coli*, *Legionella pneumophila*, *Salmonella typhimurium*, and *Vibrio cholera* [67] and other parasites such as *Leishmania amazonensis* [81]. The pathogen’s LPS and other surface antigens have been associated with non-canonical NLRP3 inflammasome activation through interaction with Guanylate binding proteins (GBPs). To our knowledge, there is no evidence of the non-canonical stimulation of the NLRP3 inflammasome by *M. tuberculosis* in pulmonary epithelial cells from transcriptomics and proteomics studies; this area remains to be investigated. However, the NLRP3 CASP4/11-dependant inflammasome is activated in macrophages during *M. tuberculosis* infection [82]. Collectively, these studies indicate that *M. tuberculosis* stimulate alveolar macrophages canonical and non-canonical NLRP3 inflammasome, leading to production of inflammatory cytokines, IL-1β, and IL18 in a TLR4-NF-kB dependent manner. Furthermore, despite increased expression of the NLRP3 inflammasome transcripts and subsequence IL1 cytokine production in pulmonary epithelial cells, the presence of this complex and other types of inflammasomes such as NLRP6, NLRC4, and AIM2 remains to be investigated and confirmed in future studies. The activation of a canonical and non-canonical alveolar NLRP3 molecular mechanism during *M. tuberculosis* infection is depicted in Figure 1.

It is apparent that alveolar macrophages produce IL1 cytokines [72,75,76,82,83] through the activation of canonical and/or non-canonical NLRP3 inflammasome by the TLR-MYD88 signalling pathway, which results in stimulation of NF-κB transcriptional factors that transcribe pro-inflammatory cytokines during *M. tuberculosis* infections. However, this mechanism remains to be confirmed for pulmonary epithelial cells. IL1 are among the group of pro-inflammatory cytokines that are essential in host defence against *M. tuberculosis* [77]. Mice with IL1α and IL1β double knockouts and IL1R type I-deficient mice display a defective granuloma phenotype accompanied with increased mycobacterial growth [79]. IL18 is known as an interferon-γ inducing factor, which has a vital role in T helper1 (Th1) response [84,85] and has been shown to be responsible for the production of pro-inflammatory cytokines, chemokines, and transcriptional factors [86,87]. Moreover, IL18 defective mice were shown to be highly susceptible to *M. tuberculosis* and BCG strains [88]. Despite the critical role played by IL1 cytokines of the NLRP3 inflammasome, overproduction of pro-inflammatory cytokines without efficient anti-inflammatory response has been associated with increase disease severity and high bacterial burden [89] that can compromise effective host response during TB infection. The presence of IL1β was associated with caseous granulomatous inflammation during *M. tuberculosis*, while blocking IL1β production relieved pulmonary inflammation [90]. Significantly high IL1β and IL18 was identified in drug resistant compared to drug susceptible TB [91], while high cytokine concentrations were identified in patients with a severe disease [92]. Therefore, the ability of pulmonary epithelial cells and alveolar macrophages to produce IL1 (IL1β and IL18) cytokines through the activation of the inflammasome may contribute to the inflammatory response during early infection by *M. tuberculosis* that is pathological to the host, thus novel strategies must be exploited to regulate this response.

## 4. Medicinal Plants and Their Bioactive Derivatives as Regulators of Alveolar NLRP3 Inflammasome during *M. tuberculosis* Infection

The high prevalence of TB worldwide is partly due to development of drug resistant strains, rendering the current treatment regimen ineffective [93,94,95]. Several studies have proposed the use of medicinal plants or their bioactive compounds with antimycobacterial activity [29,33,96,97,98,99]. However, recent studies have also exploited the use of medicinal plants and/or their derivatives that are regulators of the immune system during *M. tuberculosis* infection [100,101,102]. Medicinal plants and their respective bioactive compounds that regulate the components of the immune system may be a promising strategy in controlling Mycobacterial infections because they are not directly targeting the bacillus, reducing the pathogen’s need to develop drug resistance and selective pressure [103]. Therefore, medicinal plants and their bioactive derivatives may provide a novel host-directed perspective in controlling immunopathology in the alveoli through inhibition of the NLRP3 inflammasomes pathways that are crucial in the production of IL1β and IL18 cytokines during *M. tuberculosis* infections. Medicinal plants and/or their bioactive derivatives that are used to inhibit NLRP3 inflammasome gene transcription, formation of the inflammasome complex and production of NLRP3 inflammasome proteins (including IL1 cytokines) are reviewed below.

### 4.1. NLRP3 Inflammasome Transcript Inhibitors

*Michelia compressa* and *Michelia champaca* plants from the Magnolia family [104] possess the natural guaianolide sesquiterpene lactone, micheliolide, with a potent anti-inflammatory activity [105,106,107,108]. Zhang et al. [31] used an in vitro mouse macrophage-like cell line to investigate the anti-inflammatory activity of micheliolide during *M. tuberculosis* infection. The NF-κB transcriptional factor and NLRP3 inflammasome that are essential for transcription of IL1β, IL18, and pro-caspase 11 transcripts were downregulated, accompanied by reduction of inflammatory cytokines (IL1β and TNF-α) in a dose-dependent manner. This study concluded that the micheliolide downregulates the activation of NLRP3 inflammasome by modulating the inflammatory response induced by *M. tuberculosis* through the PI3K/Akt/NF-κB pathway.

NLRP3 inflammasome-specific transcript can be inhibited through the adaptor molecules that signal to activate transcriptional factor, NF-κB. Guttiferone K is the active compound isolated from *Garcinia yunnanensis* Hu plant [109] and has anti-cancer activities, including inhibition of autophagy and metastasis while promoting apoptosis of cancer [110]. The anti-inflammatory activity of Guttiferone K was observed through phosphorylation inhibition of the NF-κB transcriptional factor by Interleukin-1 receptor associated kinase (IRAK1) in the TLR signalling pathway of the *M. tuberculosis* H37Ra (avirulent) infected macrophages. This inhibition resulted in significantly reduced TLR-NF-κB inflammatory mediators, IL1β, TNF-α, IL6, inducible nitric oxide synthase (iNOS), and cyclooxygenase-2 (COX-2) [32]. The Guttiferone K anti-inflammatory molecular mechanism remains to be investigated in the virulent strains of *M. tuberculosis*.

There are many medicinal plant bioactive derivatives that act as NLRP3 inflammasome transcript inhibitors identified in non-*M. tuberculosis* models. These include Arctigenin [111], Silymarin [112], Rutin [113], Genistein [114], Aloe emodin [115], Anemoside B4 [116], epigallocathechin-3-gallate [117], Resveratrol [118], Silibinin [119], Artemisinin [120], Icariin [121], Polydatin [122], Cinnamaldehyde [123], Huangkui capsule [124] whose administration results in the inhibition of the inflammasome-associated transcription on both in vitro and in vivo models (Figure 2). It is crucial that these medicinal plant compounds be tested in *M. tuberculosis* in vitro and in vivo models to identify their potential use as host-directed immunomodulatory plant derivatives that can be used to prevent lung pathology during infection.

### 4.2. NLRP3 Inflammasome Protein Inhibitors

Medicinal plants and their derivatives can exert their anti-inflammatory activity in the NLRP3 inflammasome by directly inhibiting production of proteins that regulate this inflammatory pathway, including production of IL1 cytokines. The anti-inflammatory activity of andrographolide, the labdane diterpenoid isolated in the stems, and leaves of *Andrographis paniculate* was observed in macrophages co-cultured with pulmonary epithelial cells that had significantly reduced IL1β cytokine production during *M. tuberculosis* infection. This IL1β reduction in macrophages resulted in declining chemokine (IL8 and MCP-1) expression in neighbouring pulmonary epithelial cells. The proposed molecular mechanism behind IL1β inhibition is the activation of macrophage autophagy to degrade NLRP3, thus inhibiting the inflammasome activation and IL1β production. Moreover, andrographolide downregulated the phosphorylation of Akt/mTOR and NF-𝜅B p65 subunit, which will interfere with the NLRP3 inflammasome transcript expression that is essential for the maturation of the inflammasome complex [35]. Andrographolide is a natural labdane diterpenoid that was isolated in *Andrographis paniculate* plant [125] with many biological activities, including anti-cancer, antimicrobial, anti-diabetic, anti-inflammatory, etc. [126].

Several in vitro and in vivo models have identified other medicinal plant derivatives that have been found to inhibit NLRP3 inflammasome proteins and subsequent IL1 production; and these include Ginsenosides (Retinoblastoma and Compound K) [127,128], Curcumin [129], Genipin [130], Mangiferin [131], Salvianolate [132] that were derived from *Panax ginseng*, *Curcuma longa*, *Gardenia jasminoides*, *Mangifera indica*, *Salvia miltiorrhiza* medicinal plants, respectively.

### 4.3. Inhibitors of the NLRP3 Inflammasome Complex

Several medicinal plants and/or their bioactive derivatives can interfere with the formation of the inflammasome complex or the components that are needed for a complete NLRP3 inflammasome to facilitate maturation of the IL1 cytokines. The flavonoid Baicalin, isolated from *Scutellaria baicalensis* possess an anti-NLRP3 inflammasome activity during *M. tuberculosis* through decreased phosphorylated protein kinase B and mammalian target of rapamycin, decreasing NLRP3 inflammasome and subsequently IL1β production [34]. The COX-2 regulates the activation of the NLRP3 inflammasome through increased expression of LPS-induced pro-IL1β and NLRP3 by NF-κB [133]. The crude petroleum ether, ethanol, dichloromethane extracts of South African medicinal plants, *Abrus precatorius subsp. Africanus*, *Ficus sur*, *Pentanisia prunelloides*, and *Terminalia phanerophlebia* were shown to inhibit the COX-2 enzyme, thus may contribute to the anti-NLRP3 inflammasome response during *M. tuberculosis* infection [33]. The bioactive compound in these plants and their possible molecular mechanism against the NLRP3 inflammasome remains to be characterised.

NLRP3 inflammasome complex inhibitors in other models include Sweroside [134], Oridonin [135], Isoliquiritigenin [136], Cardamonin [137], Ginsenosides (Rg3, Rg1, 20S-protopanaxatriol, 25-OCH3-PPD) [138,139,140,141], Triptolide [142], Glycyrrhizin [143], Saikosaponin A [144], Quercetin [145], Dihydroquercetin [146], Casticin [147], Phloretin [148], Pterostilbene [149], Apocynin [150], Lycorine [151], Matrine [152], Tetramethylpyrazine [153], Astaxanthin [154], Danggui Buxue Tang [155], *Quamoclit angulate* plant [156], 4- Sulforaphane (methylsulfnylbutyl isothiocyanate) [157], Obovatol [158], Berberine [159] (Figure 2). Both in vitro and in vivo *M. tuberculosis* models exploring these bioactive plant derivatives remain to be investigated for their potential use as regulators of the alveolar NLRP3 inflammasome.

There is growing appreciation of medicinal plants and their bioactive molecules in management of inflammatory response regulated by the NLRP3 inflammasome as excellently reviewed by several authors [160,161,162,163]. To date, only five studies [31,32,33,34,35] have explored the use of medicinal plants and/or their bioactive derivatives in the management of the inflammasome in *M. tuberculosis* infection models. There is a growing understanding that *M. tuberculosis* infection is a multifaceted condition that is also driven by the host inflammatory response. Thus, controlling the alveolar NLRP3 inflammasome seem to be one of the novel strategies for host directed control measures of pathological response induced by *M. tuberculosis*.

## 5. Conclusions

The activation of the NLRP3 inflammasome in alveolar macrophages and potentially in pulmonary epithelial cells is an essential response of innate cells to *M. tuberculosis* that results in the production of IL1 inflammatory cytokines. Overstimulation of the alveolar NLRP3 inflammasome may contribute to the exaggerated inflammatory response that is pathological to the host. Recently, several studies have shown that medicinal plants and/or their derivatives, Andrographolide, Baicalin, Micheliolide, Guttiferone K, *Abrus precatorius subsp. Africanus*, *Ficus sur*, *Pentanisia prunelloides*, and *Terminalia phanerophlebia* can inhibit the NLRP3 inflammasome at transcriptional and post-transcriptional levels in *M. tuberculosis* models. Therefore, these plants and bioactive derivatives can be explored for anti-TB host directed adjuvants to control the *M. tuberculosis*-induced inflammatory response. Furthermore, many studies have identified many regulators of the NLRP3 inflammasome that remains to be tested on both in vitro and in vivo *M. tuberculosis* models. Future studies should investigate the ideal route of administration of these medicinal plants and their bioactive derivatives due to challenges associated with oral availability of medicinal drugs. Exploring alternative strategies such as using medicinal plants and their bioactive derivatives to regulate the host inflammatory response in TB may bring us closer to winning the battle against *M. tuberculosis*.

## Figures and Tables

**Figure 1 ijms-22-09497-f001:**
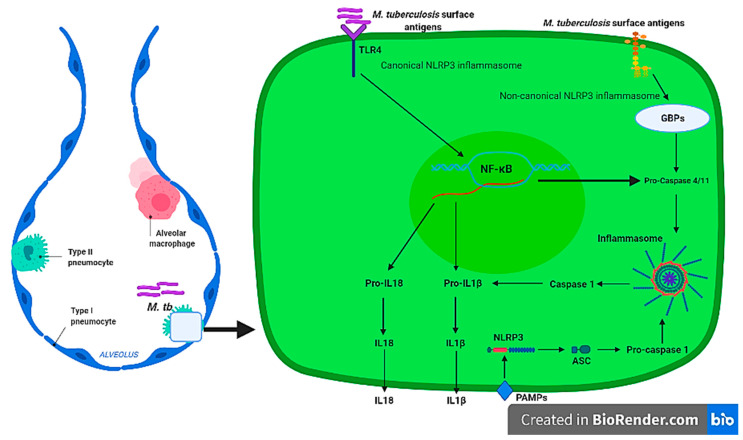
Overview of the alveolar NLRP3 inflammasome activation on alveolar innate cells during *M. tuberculosis* infection. The host PRRs (TLR4) is activated by *M. tuberculosis* surface PAMPs stimulate the NF-kB transcriptional factor to transcribe *IL1*, *IL18*, and *CASP4/11* that are processed by activation of the inflammasome complex through NLRP3 and CASP1. Image was created in BioRender.com (accessed on 23 August 2021).

**Figure 2 ijms-22-09497-f002:**
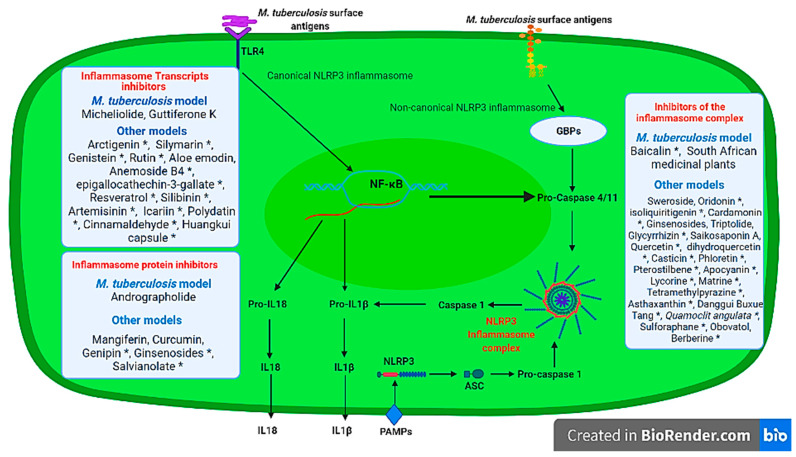
Medicinal plant and/or bioactive regulators of the NLRP3 inflammasome that can be exploited as immunomodulatory host-directed adjuvants during *M. tuberculosis* infection. Compounds denoted with * were tested in vivo. Image was created in BioRender.com (accessed on 23 August 2021).

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
