# Peer review of "Exploring the Use of Medicinal Plants and Their Bioactive Derivatives as Alveolar NLRP3 Inflammasome Regulators during Mycobacterium tuberculosis Infection"

_ijms, 2021, doi:10.3390/ijms22179497_

Round 1

Reviewer 1 Report

In this review, the authors present how the bioactive compounds of medicinal plants inhibit the NLRP3 inflammasome following Mycobacterium tuberculosis infection. After an introduction, the authors describe how M tuberculosis interacts with alveolar macrophages and pulmonary epithelial cells. Next, they explain how the bacterial infection activates NLRP3 inflammasome and finally they summarize the bioactive compounds of medicinal plants that inhibit the NLRP3 inflammasome following M tuberculosis infection.

This review is quite precise, complete and comprehensive. The figures are also very clear.

I have however some concerns.

First, it is well known that NLRP3 inflammasome are found within the macrophages and activated after bacterial infection but is it really true in pulmonary epithelial cells? I mean, is NLRP3 really expressed next NLRP3 inflammasome formed in pulmonary epithelial cells? Was it already reported? Line 215, the authors mention an increased regulation of NLRP3, ASC and caspase-1 only after infection with clinical strains of M. tuberculosis but it does not necessarily mean that it leads to NLRP3 inflammasome activation. Moreover, other inflammasomes (like NLRP6, NLRC4 or AIM2 inflammasome) that also promote caspase-1 activation and maturation of IL-1b and IL-18 are likely involved in pulmonary epithelial cells so that bioactive compounds of medicinal plants may also inhibit these inflammasomes.

Line 187, the authors wrote that “The NLRP3 inflammasomes are classified as canonical and non-canonical, depending on the activation of Caspase 1 or Caspase 11, respectively”. Actually, NLRP3 inflammasome only lead to caspase-1 activation. This inflammasome can be activated in a non-canonical manner downstream of caspase-11 activation in mouse (caspase-4 and -5 in human) which cleaves gasdermin D, next the N-term of gasdermin D forms pores in the plasma membrane triggering pyroptosis and the release of the mature pro-inflammatory cytokines IL-1b and IL-18. Pyroptosis is linked to potassium efflux, a common trigger for NLRP3 inflammasome.

Line 193. I thought that unlike pro-IL-1b, pro-IL-18 was constitutively present within cells.

Author Response

Dear Reviewer 

Kind Regards

Dr. Mvubu

Reviewer 2 Report

The review is focused on a very interesting, complex, and rapidly evolving subject. The article is well writen and according to my knowledge give a complete overview of the situation.

Figures are well presented with a good level of precision. The only remaining question is the lack of information on the in vivo and in vitro testing of the compounds. May I suggest to add this point with an * or example beside the name of the tested compound in vivo…?

The conclusion is very honest. May I suggest a comment about the possible use of the compound as a local administration as the major problem for natural drug is the oral availability?   

At the bibliography level Line 232 may I suggest adding one article focused on canonical pathways?  

Changhoon Oh, Ambika Verma and Youssef Aachoui ; Front. Immunol., 21 August 2020 | https://doi.org/10.3389/fimmu.2020.01895 : Caspase-11 Non-canonical Inflammasomes in the Lung

Author Response

Dear Reviewer 

Kind Regards

Dr. Mvubu

Round 2

Reviewer 1 Report

My concerns have been addressed, I therefore recommend acceptance for publication.